# Systemic Photoprotection in Melanoma and Non-Melanoma Skin Cancer

**DOI:** 10.3390/biom13071067

**Published:** 2023-07-02

**Authors:** Mariafrancesca Hyeraci, Elena Sofia Papanikolau, Marta Grimaldi, Francesco Ricci, Sabatino Pallotta, Rosanna Monetta, Ylenia Aura Minafò, Giovanni Di Lella, Giovanna Galdo, Damiano Abeni, Luca Fania, Elena Dellambra

**Affiliations:** 1IDI-IRCCS, Dermatological Research Hospital, Via dei Monti di Creta 104, 00167 Rome, Italy; es.papanikolaou@gmail.com (E.S.P.); f.ricci@idi.it (F.R.); s.pallotta@idi.it (S.P.); ros.monetta88@gmail.com (R.M.); y.minafo@idi.it (Y.A.M.); g.dilella@idi.it (G.D.L.); d.abeni@idi.it (D.A.); l.fania@idi.it (L.F.); e.dellambra@idi.it (E.D.); 2Department of Pharmaceutical and Pharmacological Sciences, University of Padua, 35131Padua, Italy; 3Department of Medical and Surgical Sciences, Division of Dermatology, Catholic University of the Sacred Heart, Fondazione Policlinico Universitario A. Gemelli IRCCS, 00168 Rome, Italy; martagrimaldi64@gmail.com; 4Dermatology Unit, AORN San Giuseppe Moscati, 83100 Avellino, Italy; giovannagaldo@libero.it

**Keywords:** photoprotection, melanoma, non-melanoma, basal cell carcinoma, squamous cell carcinoma

## Abstract

Non-melanoma skin cancers (NMSCs), which include basal cell carcinoma (BCC), squamous cell carcinoma (SCC), and actinic keratosis (AK), are the most common cancer diseases in the Caucasian race. If diagnosed late and improperly treated, BCC and SCC can become locally advanced and metastasize. Malignant melanoma (MM) is less frequent but more lethal than NMSC. Given the individual and social burdens of skin cancers, performing an adequate prevention is needed. Ultraviolet (UV) ray exposure is one of the main risk factors for skin cancer. Thus, the first-choice prevention strategy is represented by photoprotection that can be both topical and systemic. The latter consists of the oral administration of molecules which protect human skin against the damaging effects of UV rays, acting through antioxidant, anti-inflammatory, or immunomodulator mechanisms. Although several compounds are commonly used for photoprotection, only a few molecules have demonstrated their effectiveness in clinical trials and have been included in international guidelines for NMSC prevention (i.e., nicotinamide and retinoids). Moreover, none of them have been demonstrated as able to prevent MM. Clinical and preclinical data regarding the most common compounds used for systemic photoprotection are reported in this review, with a focus on the main mechanisms involved in their photoprotective properties.

## 1. Introduction

Non-melanoma skin cancers (NMSCs) account for approximately 20% of all malignancies and their incidence is steadily increasing [1,2]. These tumors consist of basal cell carcinoma (BCC), squamous cell carcinoma (SCC), and actinic keratosis (AK). The reported frequency of NMSCs is underestimated because they are often not recorded in national registries and their incidence is much higher than that of malignant melanoma (MM). BCC is the most frequent carcinoma in the Caucasian race and the estimated BCC/SCC ratio is about 2.5:1 in the general population [3]. Notably, this ratio is reversed in organ transplant patients [4]. In Italy, the reported incidence of NMSCs is approximately 100 cases per 100,000 inhabitants per year, which clearly underestimates the true incidence of these tumors [4,5,6]. Both BCC and SCC, if diagnosed early and correctly treated, are completely cured in most cases; however, they can become locally advanced, and mainly SCC can metastasize, generally in proximal lymph nodes [2,7]. AKs are keratinocyte neoplasms generally occurring on photo-exposed damaged areas; if untreated, they may evolve into SCCs [2]. 

MM accounts for <5% of total diagnosed skin cancers, but it accounts for most deaths due to skin cancer. On average, in the United States, one person every hour dies from MM. Depending on histological and clinical features, MM can be classified in three major groups: superficial spreading melanoma (SSM), nodular melanoma (NMM), and lentigo maligna melanoma (LMM) [8]. 

Although both MM and NMSC are considered multi-factorial diseases, prolonged and unprotected natural exposure ultraviolet (UV) rays or UV lamps are recognized as the main causes of skin cancer. In fact, NMSCs usually occur on chronically exposed skin areas such as the head and neck region, and the dorsum of the hands and the trunk, generally in patients who perform outdoor work or hobbies (e.g., farmers, sailors, etc.). The risk of developing these tumors is significantly increased for people with fair skin [2,7]. Several types of MM, e.g., lentigo maligna melanoma, are recognized as related to prolonged UV photo-exposure. However, acute and chronic sun exposure is a risk factor also for superficial spreading melanoma and nodular melanoma.

Due to their high incidence, skin cancers represent a significant burden for the National Health System. Although these cancers have a high healing rate if detected early and well-treated, given their individual and collective burden it is imperative to perform adequate prevention [2,7]. Besides avoiding or reducing UV exposure, the first-line prevention strategy for MM and NMSCs is represented by photoprotection, which can be topical or systemic. Topical photoprotection consists of the use of sunscreens applied to the skin. Although fundamental, this approach presents some drawbacks, such as a short half-life, low patient compliance in the correct use of sunscreens, low systemic efficacy, and risk of contact dermatitis, even if this occurrence is very rare [9]. 

Systemic photoprotection consists of the oral administration of specific substances with photoprotective and anti-photocarcinogenic properties, such as nicotinamide, vitamins, minerals, polyphenols, carotenoids, and other antioxidants. These substances increase the natural protection of the body against the damaging effects of UV light and prevent carcinogenesis and photo-induced aging [9,10,11]. The photoprotectors act through many different mechanisms, including antioxidant, anti-inflammatory, or immunomodulator mechanisms [9,10,12]. Indeed, UV light induces DNA damage, increased oxidative stress, and immunosuppression of the skin [12,13,14]. Systemic photoprotection must be always accompanied by the topical one because of the higher efficacy of the latter.

The aim of this review is to summarize the state of the art of knowledge about systemic photoprotection focused on MM and NMSC prevention.

## 2. Materials and Methods

### 2.1. Literature Search

This review was prepared according to the Preferred Reporting Items for Systematic Reviews and Meta-Analyses (PRISMA) statement. We systematically searched peer-reviewed original articles published in English in the following databases: PubMed, Scopus, and Web of Science. No restriction in terms of time have been set retrospectively. The last search was performed on 21 December 2022. To perform the literature analysis, we combined the terms: “Non-Melanoma Skin Cancer*”, “Melanoma”, or “chemoprevention”, with the terms: “Vitamin D”, “Nicotinamide”, “Polypodium”, “Retinol”, “Retinoids”, “Carotenoids”, or “Celecoxib” using the Boolean connector “and” (i.e., (Non-Melanoma Skin Cancer*) AND (Vitamin D)). Secondly, we assessed the abstracts of all potentially pertinent articles to check if they met the eligibility criteria. 

### 2.2. Eligibility Criteria

In this review we included all the cross-sectional, case-control, or cohort studies concerning the use of the indicated drugs in skin cancer treatment. We also included preclinical in vitro and in vivo studies performed using the indicated drugs for the treatment of skin cells or animal models suitable for skin cancer study. The databases selected to perform this work did not report grey literature; for this reason, unpublished studies were not included in this review. The data found in conference papers or abstract books have not been reported as well, because these are usually reported later in published articles.

## 3. Vitamin D

Vitamin D (vit D) is a fat-soluble vitamin that exists in two natural forms: ergocalciferol (vitamin D2) and cholecalciferol (vitamin D3). The main natural source of this vitamin is the synthesis of cholecalciferol in the lower layers of the epidermis through a chemical reaction dependent on UVB radiation exposure. Ergocalciferol and cholecalciferol can also be taken from some foods (e.g., fish, dairy products, and cereal products) and dietary supplements. Vit D from skin synthesis or from diet is biologically inactive and is converted into the active form by two hydroxylation steps in the liver (25-hydroxyvitamin D (25(OH)D) and in the kidney (1,25-Dihydroxyvitamin D3 or 1,25-(OH)2D3). Vit D has a significant role in calcium homeostasis and regulates the calcium and phosphorous levels in the blood [15].

Preclinical studies showed that the active form of vit D (1,25-dihydroxyvitamin D3) suppressed the growth of MM and cutaneous carcinoma cells both in in vitro cultures and in animal models modulating the expression of cell cycle regulators (Figure 1) [16]. The 1,25-dihydroxyvitamin D3 protected the keratinocytes from UV radiation-induced apoptosis, reactive oxygen species (ROS) generation, and cyclobutane pyrimidine dimer (CPD) formation [17,18]. Although experimental studies indicated that vit D may have a protective effect on skin cancer risk, epidemiological studies investigating the influence on skin cancer risk of 25(OH) vit D serum level, a biomarker of vit D status, obtained conflicting results. Most studies reported a positive association between 25(OH) vit D levels and skin cancer risk [19,20]. Vojdeman et al. enrolled 217.244 individuals with a median vit D level of 46 nmol/L (IQR 27–67 nmol/L) and they identified significant positive associations between levels of 25(OH) vit D and both MM and NMSCs [21]. In a case-control study, patients with a recent diagnosis of NMSC had significantly higher mean serum levels of 25(OH) vit D, even if most patients (65.9%) had a vit D deficiency (25(OH) vit D < 30 ng/mL). However, no association was found between histological type, time from diagnosis, or a previous skin tumor history and circulating 25(OH)D levels [22]. Some studies referred to an inverse [23,24] or no association between 25(OH) vit D levels and skin cancer risk [25,26]. Significantly reduced serum levels of 25(OH) vit D were found in stage IV MM patients as compared to those at stage I. Moreover, patients with the lowest 25(OH) vit D serum levels (<10 ng/mL) developed earlier distant metastasis, suggesting a possible role of its levels in the pathogenesis and progression of MM [27]. However, a meta-analysis of 20 studies found no association between the serum levels of 25(OH) vit D and the risk of MM or NMSC [28]. 

Moreover, the clinical studies concerning the associations between dietary vit D and the risk of skin cancer yielded inconsistent results [29,30]. Indeed, some studies suggested that a high intake of dietary vit D may protect against skin cancer development [24,31], whereas other studies observed no statistically significant association [32]. A randomized controlled trial involving 36.282 participants evaluated the impact of daily calcium (1000 mg) and vit D3 (400 IU) on the incidence of NMSC and MM over a 7-year period. The results of the study showed that calcium and vit D intake had no effect on the incidence of MM and NMSCs. However, in women with a clinical history of NMSC, this supplementation reduced MM risk, suggesting a potential chemopreventive role for calcium and vit D administration in this high-risk group [33]. Although rare, toxicity may occur following vit D intake. Symptoms are related to hypercalcemia and are observed after intake of excessively high doses of vit D (50,000–2,604,000 IU/day). No side effects were reported in the studies described, because vit D intake was contained within the recommended limits [34].

A recent systematic review and dose–response meta-analysis of 13 prospective studies indicated that the circulating level of 25(OH) vit D was associated with higher risks of MM and NMSC. On the contrary, dietary, supplemental, and total intakes of vit D were not associated with MM and SCC risks. However, vit D intake was associated with a slightly increased BCC risk [35]. 

The results described here are summarized in Table 1.

## 4. Nicotinamide

Nicotinamide (or niacinamide, NAM) and Nicotinic acid (or niacin, NA) are two forms of vitamin B3, a water-soluble vitamin. NAM and NA represent the two main precursors of nicotinamide adenine dinucleotide (NAD), a key enzymatic cofactor for cellular energy [36]. However, these two molecules act on distinct metabolic pathways and have different pharmacological activities. 

Intracellular NAD acts as a cofactor regulating the oxidative/reductive activity of approximately 500 enzymes involved in cellular metabolism. NAD mediates electron transfer by oscillating between its oxidized (NAD+) and reduced (NADH) states [37]. However, NAD+ also acts as a substrate for several enzymes including Poly ADP-ribose polymerase (PARP) (Figure 1). Importantly, PARP is a nuclear enzyme that is activated in response to DNA damage to promote and coordinate DNA repair processes [38].

DNA photodamage and UV-induced immune suppression have a key role in the development of MM as well as NMSC. NAM displayed photoprotective properties in vitro and in vivo as it prevented UV-induced ATP depletion (Figure 1), restoring cellular energy and enhancing DNA repair activity in keratinocytes. Moreover, NAM modulates inflammatory cytokine production. When topically or orally administered, NAM prevented the immune suppressive effects of UV radiation and modulated the skin barrier function in mouse models [10,39]. A preclinical study showed the anti-MM activity of NAM both in vitro and in vivo [40].

Patients with a history of skin cancer are more susceptible to UV-immunosuppression. Interestingly, administration of oral NAM at doses from 500 mg once daily to 500 mg twice daily for 7 days was effective in reducing UV immunosuppression in placebo-controlled studies [41]. The incidence of NMSC increased 50–80-fold and that of MM was twice more frequent in chronically immune-suppressed transplant recipients. The results of two studies in a small number of transplant patients suggest that NAM administration may contribute to a substantial decrease in the size and number of AKs compared with a placebo [42,43]. 

Two double-blind, randomized, placebo-controlled phase 2 trials evaluated the effects of oral NAM at doses of 500 mg twice daily and 500 mg once daily in patients with high AK predisposition. After 4 months of treatment, NAM was able to significantly reduce AK recurrence and this reduction was greater in patients treated with NAM 500 mg twice daily (35%) than in patients treated with 500 mg/day (29%) [44].

A phase 3 randomized controlled trial evaluated the effects of the oral intake of NAM at 1 g/day for 12 months in patients at high risk for NMSC. The study showed a 23% rate reduction of new NMSC in patients receiving NAM. However, there is no evidence of benefits after the 6-month discontinuation of NAM administration [45]. No difference in MM chemoprevention was observed with NAM administration compared to the control. However, the not eligible criteria excluded patients with a history of melanoma over the last 5 years, since the trial was designed to study the efficacy in NMSCs. Therefore, larger phase 3 studies could be performed to assess the safety and efficacy of NAM chemoprevention in patients at high MM risk.

A recent systematic review and meta-analysis of randomized controlled trials showed that NAM was associated with a significant reduction in BCCs and SCCs, and increased risk of gastrointestinal adverse effects (i.e., diarrhea) in high-risk skin cancer patients or organ transplant recipients [46].

Nicotinamide toxicity has been reported at doses greater than 3.5 g/day [47]. In Europe, the recommended upper limit of nicotinamide is 900 mg/day, whereas for niacin it is 54 mg/day [46]. 

Nicotinamide was mentioned in the NCCN Guidelines for the prevention of NMSC in high-risk patients (NCCN Guidelines for Patients Squamous Cell Skin Cancer). In detail, it is reported as an agent which may prevent recurrence and/or spreading of SCC to distant sites in high-risk patients [48].

As regards the potential effectiveness of nicotinamide in the prevention of MM, results reported in an interesting preclinical study suggested anti-melanoma activity both in vitro and in vivo even if at high doses (above 20 mM), giving the basis for further future investigations on a potential role for nicotinamide in MM management [40]. Nevertheless, no significant evidence of effectiveness of nicotinamide in prevention of MM was highlighted by any clinical study, as reported in a recent systematic review and meta-analysis [46].

The results described here are summarized in Table 1.

## 5. *Polypodium leucotomos*

*Polypodium leucotomos* (PL) is a tropical species of fern from South and Central America, where it has been used in traditional medicine as a therapeutic agent for psoriasis [49,50]. PL extract (PLE) obtained from the leaves of the fern has a high phenolic content (benzoates and cinnamates). Several in vitro studies were carried out to test the direct antioxidant and photoprotective effects of PLE (Figure 1) [51,52,53,54]. A group of researchers found that PLE has a protective effect on a human keratinocytes cell line (HaCaT), preventing both the cytotoxic damage and apoptosis induced by UV exposure [55], as well as inhibiting Nitric Oxide (NO) and Tumor Necrosis Factor (TNFα) production and inducible NO synthase (iNOS) upregulation after UV-exposure [53].

Several studies about PLE were performed on healthy human volunteers. In 1996, Gonzalez S et al. reported a significant reduction in UV-induced erythema, compared to the control group, in individuals pre-treated with the topical application of PLE, suggesting an interesting PLE photoprotective activity [52]. Moreover, a significant reduction in CPDs (cyclobutane pyrimidine dimers), sunburns, and proliferating cells, compared to the control group, was detected in specimens from persons receiving oral administrations of PLE [56]. In 2015, a clinical trial was carried out to test the efficacy of PLE for the treatment of AKs: 34 bald men with at least two AKs on the scalp were enrolled and divided into two groups. All participants underwent two sessions (one per week) of methyl aminolevulinate (MAL) photodynamic therapy (PDT), and one group also received oral supplementation with PLE one week after the PDT session. Results showed that PLE improved the effects of PDT on AKs and reduced AK recurrence [57]. Additional studies evaluated the role of PLE in specific dermatoses and/or skin conditions. Two studies showed subjective improvement in idiopathic photodermatoses after two weeks of treatment with PLE [58,59] while the use of PLE in melasma yielded mixed results [60,61].

Although the possible mechanisms of action of PLE have not yet been fully elucidated, these studies indicate that PLE may be a useful addition to photoprotection; however, to date there are no clinical trials confirming its role in chemoprevention.

The results described here are summarized in Table 1.

## 6. Retinol and Retinoids

Retinol and retinoids are natural forms or synthetic derivatives (i.e., acitretin, isotretinoin, etretinate) of vitamin A that exert a key role in the regulation of many important biological processes. In particular, retinoids regulate epithelial maturation, cellular differentiation and proliferation, growth arrest, and apoptosis by activating nuclear retinoid receptors (RAR or RXR), which work as ligand-dependent transcriptional factors. The interaction with the retinoid leads to the formation of a complex (RARE) between the RAR/RXR heterodimer and the ligand (Figure 1) [62]. 

Retinoids were used for the chemoprevention of some cancers, i.e., bladder cancer and second malignancies in the liver and in the breast, with promising results [63,64].

Retinoids were mentioned in the NCCN Guidelines for the prevention of NMSCs in patients with a high risk of developing NMSCs. In particular, oral administration of acitretin or isotretinoin is reported as a useful agent in chemoprevention of SCC in patients who develop AK easily [48]. Retinoids were evaluated in prospective studies in patients with a high risk of NMSCs, including organ transplant recipients [65,66,67,68,69,70], patients affected by xeroderma pigmentosum [71], or psoriatic patients treated with PUVA therapy (psoralen plus UV-A) [72]. The results of these studies indicated that the oral administration of acitretin [65,66,67,69], etretinate [68], or isotretinoin [71], was able to significantly reduce the incidence of new NMSCs compared to placebos in patients with high risk for multiple neoplastic lesions [65,66,67,68,69,70,71,72]. However, the efficacy of oral retinoids in patients with a moderate risk of developing NMSCs appeared less convincing. Only one randomized controlled trial was conducted in patients with ten or more previous AKs (at least one AKs in the past year) but no more than two SCCs/BCCs. Results showed a reduction in the risk of developing additional SCCs but not BCCs in patients who had taken oral retinol compared with a placebo [73]. Nevertheless, the scarce available literature is insufficient to fully understand the possible usefulness of retinoids in these patients. 

It has to be recognized that side effects associated with oral retinoids may be significant. The most common adverse events reported in the literature are cheilitis, dryness and excessive peeling of the skin, and hair disorders [65]. Less frequent side effects include, among others, headache, epistaxis, hepatic and lipid changes, osteoporosis, calcification of the ligaments, and neurological disorders [66,71,74]. Notably, in a large, randomized trial that was carried out on subjects with resected stage I non-small cell lung cancer, the use of isotretinoin has been related to increased mortality [75]. To limit these adverse effects, one study evaluated topical administration of treinoin 0.1% compared with a control vehicle in patients at risk of developing NMSCs. However, no significant difference among the treatment arms for the development of new invasive or in situ SCC, BCC, or AKs was observed [76]. 

Although several studies have provided results supporting the effectiveness of retinoids in the prevention of NMSCs, at least for high-risk patients, there is scarce evidence regarding their usefulness for MM prevention. The sole clinical trial available in the literature was carried out in 1994 by Halper et al. to evaluate the effects of the topical application of 0.1% tretinoin. The obtained results suggested a histological improvement in dysplastic nevi, potential precursors of MM [77]. Nevertheless, no further study was performed to test the effect of systemic administration of retinoids in chemoprevention of MM.

The results described here are summarized in Table 1.

## 7. Carotenoids

Carotenoids are molecules with antioxidant and quenching properties arising from the presence of a long-chain polyene structure and are present in all green plants. The greater the number of conjugated double bonds, the higher the quenching power of the molecule [78]. Depending on their polarity, carotenoids can be classified as carotenes (i.e., α-, β- carotene, lycopene) and xanthophylls, more polar because of the presence in the structure of hydroxy- or keto- functional groups (i.e., lutein and astaxanthin) [79].

The eligibility of carotenoids to be used in photoprotection arises from their antioxidant and quenching properties (Figure 1) and from the knowledge of their mechanisms of action: they can activate the antioxidant response element transcription system [80] and upregulate gap junctional cell–cell communication by modulating connexin gene expression [81]. The effects of carotenoids on gene expression have been explained as the result of the interaction between the metabolites of carotenoids (mainly retinoic acid) [82] with both the retinoic acid receptor (RAR) and retinoic X receptor (RXR) [83,84].

The effectiveness of carotenoids in oral photoprotection was evaluated in a randomized, placebo-controlled, double-blind clinical trial, which demonstrated the photoprotective capacity of Nutrilite™ Multi Carotene, a mixture of carotenoids. In particular, the oral intake of such a mixture exerted a protective effect on human skin against both UVA and UVB radiation by increasing the minimal persistent pigmentation dose and the minimal erythemal dose [85].

Among carotenes, β-carotene showed effectiveness in systemic photoprotection in patients affected by erythropoietic protoporphyria and with higher photosensitivity in comparison to healthy individuals [86]. In 1975, the FDA approved its use in the management of photosensitivity in patients affected by erythropoietic protoporphyria. Important limits of using β-carotene for systemic photoprotection are related to its safety profile. Indeed, its administration for long treatment periods (>10 weeks) and at high doses (>12 mg/day), required the exertion of significant therapeutic effects, and had deleterious effects on high-risk patients for lung cancer [87].

As regards the potential use of β-carotene for chemoprevention of both MM and NMSC, some encouraging preclinical results were obtained. In particular, Guruvayoorappan et al., indicated the ability of β-carotene to downregulate in a MM cell line the expression of TNF-α, a tumor growth factor, and iNOS, which synthesizes Nitric Oxide, a mediator of vasodilatation [88]. Despite this in vitro study, no clinical study has reported on the significant effectiveness of carotene for prevention of NMSCs or MM [89].

Among polar carotenoids, astaxanthin is the molecule characterized by the highest antioxidant capacity. It is one of the most common among xanthophylls, and it is mainly found in crustaceans and Salmonidae. Astaxanthin has both hydrophilic and hydrophobic properties as its structure contains both hydrophilic functional groups and conjugated double bonds. These chemical features provide this molecule with the ability to scavenge free oxygen radicals both from the cell surface and from the phospholipidic bilayer of the cell membrane (Figure 1) [90]. The antioxidant power of astaxanthin does not arise only from its scavenger activity, but also from its ability to modulate the expression of enzymes involved in ROS metabolism. Indeed, increased levels of nuclear factor erythroid 2-related factor (Nrf2) were detected in irradiated cells, pre-treated with astaxanthin. Nrf2 binds the promoter regions of antioxidative enzymes as glutathione peroxidase 1, catalase, HEME oxygenase-1, and superoxide dismutase 2 [91,92].

Such observed antioxidant properties prompted researchers to further investigate the effects of astaxanthin on humans with a main focus on its eventual photoprotective effect, being the oxidative stress one of the main mechanisms involved in UV-induced skin damage [93]. In particular, the authors of a randomized, double-blind, placebo-controlled trial found that dietary supplementation with astaxanthin enhanced the minimal erythemal dose and reduced the dryness caused by UV rays in healthy human skin [94].

Furthermore, astaxanthin was shown to act specifically on biological targets involved in tumor progression. In particular, some in vitro studies indicated that astaxanthin exerted an antiproliferative effect on tumor cell lines by regulating the expression of NF-kB, a family of transcription factors of several genes, such as growth factors and genes involved in inflammation [95] and by inhibiting key events in the JAK-2/STAT-3-signaling cascade [96]. Despite these encouraging preclinical and clinical results, no effectiveness of astaxanthin in the prevention or the management of NMSCs and MM has been actually highlighted in clinical trials. 

The results described here are summarized in Table 1.

## 8. Celecoxib

Celecoxib is an anti-inflammatory drug approved for the temporary relief of pain and the treatment of inflammation in patients affected by osteoarthritis or rheumatoid arthritis [97]. Its well-known mechanism of action consists of the selective inhibition of isoform 2 of the cyclooxygenase (COX) enzyme (Figure 1). The latter catalyzes the conversion of arachidonic acid (Ar Ac) into unstable prostaglandins G2 (PGG2), rapidly converted into PGH2, which undergo further reactions leading to PGE2 [98]. Although these molecules are mainly involved in the inflammation process, they can stimulate cell proliferation and motility, induce angiogenesis, and inhibit immune surveillance and apoptosis [99]. Interestingly, enhanced levels of COX2 have been found in both AK [100] and SCC [101]. Many growth factors, cytokines, and oncogenes induce the increase in COX2 expression. Moreover, wildtype p53, the well-known “guardian of the genome”, negatively regulates COX2 levels. On the contrary, mutant p53 is ineffective in this regard [102,103]. Therefore, targeting COX2 could result in carcinogenesis suppression. Several preclinical studies have been carried out in experimental animals. In particular, in 2001 Neufang et al. reported that transgenic mice overexpressing the gene of human COX2 in basal keratinocytes developed epidermal hyperplasia and dysplasia [104]. Furthermore, the treatment of mice with selective COX2 inhibitors resulted in a chemopreventive effect against UV-induced skin tumors [105]. Thus, clinical studies have been carried out to investigate the potential chemopreventive role of COX2 inhibitors. In particular, a double-blind placebo-controlled randomized trial was performed by Elmets et al. in 2010 on 240 patients considered at high risk for NMSC, to test the efficacy of celecoxib in NMSC chemoprevention. Celecoxib was able to reduce the incidence of BCC and SCC but not AK [106]. Despite such encouraging efficacy of the data, some interesting evaluations regarding its safety profile are worthy reporting. As a COX2 inhibitor, it can be responsible for the appearance of important cardiovascular side effects (i.e., increased blood pressure, myocardial infarction, stroke, or vascular death) [107]. Despite its interesting chemopreventive properties, the risk/benefit balance does not allow mention of the drug in international guidelines for NMSC prevention.

As regards the potential effectiveness of celecoxib in the prevention of MM, a preclinical in vitro study reported its capacity to reduce colony formation and cell motility in a panel of melanoma cell lines, without exerting any antiproliferative effect [108]. No clinical studies focusing on the potential role of celecoxib in management of MM have been carried out.

## 9. Conclusions

Several studies reported the beneficial effects of certain molecules or vitamins utilized as systemic photoprotection for NMSC; however, only a few works reported their possible role in the prevention of MM.

To date, many of these compounds are utilized as supplements to reduce UV-induced erythema (i.e., astaxantin, caroitenoids, lycopene, or polypodium extract) but none of them has demonstrated a real efficacy in clinical trial for the chemoprevention of skin cancer.

The only drugs mentioned in international guidelines for the prevention of NMSC and utilized as chemoprevention for these tumors are retinoids (i.e., acitretin and isotretinoin) and nicotinamide. Retinoids are indicated only in high-risk patients to develop NMSC (i.e., organ transplant recipients, patients with xeroderma pigmentosum, or psoriatic patients treated with PUVA therapy) but they may cause severe side effects. On the other hand, oral nicotinamide has demonstrated a chemopreventative role for NMSC in several clinical trials without inducing significant side effects.

The chemoprevention role of some supplements or drugs regarding MM is still a matter of debate because the available literature does not support any positive conclusion.

**Table 1 biomolecules-13-01067-t001:** This table summarizes the uses in skin protection, the study results, and the levels of evidence for vit D, NAM, PLE, acitretin, etretinate, isotretinoin, β-carotene, astaxanthin, and celecoxib.

Molecule	Use	Dose	Outcome	Side Effects	Level of Evidence ^1^	Refs.
Vit D	NMSC prevention	Highly variable	Insufficient data for conclusive considerations	NR	IB	[19,20,21,22,23,24,25,26]
	MM prevention	400 IU/day	No effectiveness		IB	[28]
NAM	UV-induced immunosuppression prevention	0.5–1.5 g/day	Significant reduction of UV-induced immunosuppression	Diarrhea	IB	[41]
	AK prevention		Reduction of AKs recurrence in at-risk subjects and decrease in number and size of AKs		IB	[42,43,44]
	NMSC prevention		Reduction in the rate of occurrence of new NMSC		IA	[46,48]
	MM prevention	NR	No significant effectiveness		IB	[46]
PLE	Photoprotection	480 mg/day	Reduction in UV-induced erythema	NR	IB	[52]
	AKs	480–960 mg/day	Improvement of the clearance rate of scalp AKs in patients undergoing PDT		IIA	[57]
	Idiopathic photodermatoses	480 mg/day	Subjective improvement		IIB	[58,59]
	Melasma	480–720 mg/day	Contrasting results		IB	[60,61]
Acitretin	NMSC prevention	15–50 mg/day	Reduction of NMSC incidence in high-risk patients	Dryness and peeling of the skin, hair disorders, headache, epistaxis, osteoporosis, calcification of ligaments, neurological disorders	IB	[48,65,66,67,69]
Etretinate	Skin cancer prevention	20 mg/day	Reduction of skin cancer incidence in high-risk patients		IB	[48,68]
Isotretinoin	Skin cancer prevention	150 mg/day	Reduction of skin cancer incidence in high-risk patients		IB	[48,71]
β-carotene	Photoprotection	15–180 mg/day	Improved tolerance to sun light in patients affected by erythropoietic protoporphyria	Increased incidence of lung cancer in high-risk patients	IB	[86]
Astaxanthin	Photoprotection	4 mg/day	Increased minimal erythemal dose and reduced skin dryness after UV exposure	NR	IB	[93]
Celecoxib	AK prevention	200 mg twice daily	No effect	Increased blood pressure, myocardial infarction, stroke, or vascular death	IB	[106]
	NMSC prevention		Reduction of BCC and SCC incidence		IB	[106]

^1^ Levels of evidence are based on guidelines in the Journal of the American Academy of Dermatology: level IA indicates that evidence is derived from a meta-analysis of randomized controlled trials; level IB indicates that evidence is derived from at least one randomized controlled trial; level IIA indicates that evidence is derived from at least one nonrandomized trial; level IIB indicates that evidence is derived from at least one other type of experimental study; level III data include evidence from non-experimental descriptive studies, such as comparative studies, correlation studies, and case-control studies; and level IV indicates that evidence is derived from reports or expert opinions, or the clinical experience of respected authorities, or both.

## Figures and Tables

**Figure 1 biomolecules-13-01067-f001:**
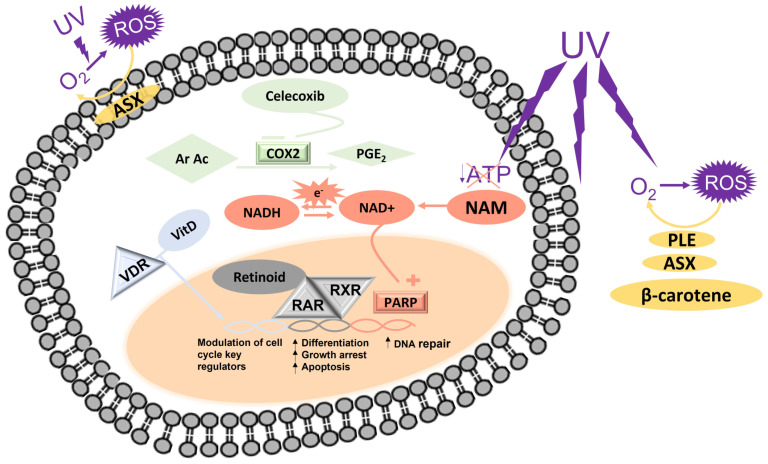
This figure details the main mechanisms involved in the chemopreventive action of the drugs included in this review. Vit D interacts with vit D receptor (VDR) leading to the formation of a complex which migrates into the nucleus, interacts with DNA, and modulates the expression of cell cycle key regulators. NAM prevents UV-induced ATP depletion and is a precursor of NAD, an enzymatic cofactor, which mediates electron transfer by oscillating between its reduced (NADH) and oxidized (NAD+) states. The latter works as a substrate for PARP, a nuclear enzyme involved in DNA repair processes. PLE, Astaxanthin (ASX), and β-carotene are antioxidant agents that neutralize ROS, leading to the restoration of molecular oxygen. Retinoids interact with retinoid receptors (RAR or RXR) in the nucleus, thus leading to the formation of RARE complex. This latter acts as transcriptional factor for genes involved in cellular differentiation, growth arrest, and apoptosis. Celecoxib is an anti-inflammatory drug, inhibitor of cyclooxygenase−2 (COX2), and a key enzyme in the synthesis of prostaglandins (PGE_2_) from arachidonic acid (Ar Ac).

## Data Availability

Data is available within the article.

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
