# Peer review of "Systemic Photoprotection in Melanoma and Non-Melanoma Skin Cancer"

_biomolecules, 2023, doi:10.3390/biom13071067_

Round 1

Reviewer 1 Report

This is an exciting and thorough review of the efficacy of orally taken molecules to prevent subtypes of skin cancer: basal cell carcinoma, squamous cell carcinoma, actinic keratosis, and the more lethal malignant melanoma. 

There are only a few minor comments:

Please correct:

Line 52 “Although both MM and NMSC are considered pub-torial diseases...

 Table 1

Significant revention of UV-induced immunosuppression

400 IU/die

0.5-1.5 g/die

480 mg/die

480-960 mg/die

480 mg/die

480-720 mg/die

 Line 309: However, no significant difference between the treatment arms for the development of...

 Line 360: “and from the phospholipidic belayer of the...

 LIne 383: for the temporarily relief of pain

 Line 374: Delete comma some in vitro studies, indicated...

 Line 402: Delete initials: Elmets CA et al....

Author Response

Reviewer #1

This is an exciting and thorough review of the efficacy of orally taken molecules to prevent subtypes of skin cancer: basal cell carcinoma, squamous cell carcinoma, actinic keratosis, and the more lethal malignant melanoma. 

There are only a few minor comments:

Please correct:

Line 52 “Although both MM and NMSC are considered pub-torial diseases...”

"Pub-torial" has been replaced by "multi-factorial".

Table 1

Significant revention of UV-induced immunosuppression

"Revention" has been replaced by "reduction"

400 IU/die

0.5-1.5 g/die

480 mg/die

480-960 mg/die

480 mg/die

480-720 mg/die

For the reported doses, “die” has been replaced by “day”

Line 309: “However, no significant difference between the treatment arms for the development of...”

"Between" has been replaced by "among"

 Line 360: “and from the phospholipidic belayer of the...”

"Belayer" has been replaced by "bilayer"

 LIne 383: “for the temporarily relief of pain”

"Temporarily" has been replaced by "temporary"

 Line 374: Delete comma “some in vitro studies, indicated...

Deleted

 Line 402: Delete initials: “Elmets CA et al....”

Deleted

Reviewer 2 Report

The review article by Hyeraci et al. is well-written and covers an important area in the biology of skin cancer. They have used adequate references to bolster their statements. With a few minor edits, this work can be published.

1) Figure 1: The review article only contains one figure which is a central figure of their paper as it is a showcase for all the major mechanisms described by the authors. It will be very helpful if authors expand the legend to better describe the figure They is a use of different colors in the figure but its importance is not clearly defined neither in the figure nor in the legend.

2) Table 1. It will be good to add another column which list the key references that authors deem fit to describe the summary outcome for each of the molecule in the table. 

3) Line 238 repetition of word "further"

4) Line 260 needs to be rephrased. 

Author Response

Reviewer 2

The review article by Hyeraci et al. is well-written and covers an important area in the biology of skin cancer. They have used adequate references to bolster their statements. With a few minor edits, this work can be published.

1) Figure 1: The review article only contains one figure which is a central figure of their paper as it is a showcase for all the major mechanisms described by the authors. It will be very helpful if authors expand the legend to better describe the figure They is a use of different colors in the figure but its importance is not clearly defined neither in the figure nor in the legend.

The legend of figure 1 was expanded for a more complete description.

2) Table 1. It will be good to add another column which list the key references that authors deem fit to describe the summary outcome for each of the molecule in the table. 

References have been provided in the table.

3) Line 238 repetition of word "further"

Actually, in line 238 the word “further” in not repeated.

4) Line 260 needs to be rephrased. 

The indicated line has been rephrased